# The Effect of Zearalenone on the Cytokine Environment, Oxidoreductive Balance and Metabolism in Porcine Ileal Peyer’s Patches

**DOI:** 10.3390/toxins12060350

**Published:** 2020-05-27

**Authors:** Kazimierz Obremski, Wojciech Trybowski, Paweł Wojtacha, Magdalena Gajęcka, Józef Tyburski, Łukasz Zielonka

**Affiliations:** 1Department of Veterinary Prevention and Feed Hygiene, Faculty of Veterinary Medicine, University of Warmia and Mazury in Olsztyn, Oczapowskiego 13/29, 10-718 Olsztyn, Poland; mgaja@uwm.edu.pl (M.G.); lukaszz@uwm.edu.pl (Ł.Z.); 2Regional Veterinary Inspectorate in Gdańsk, Na Stoku 50, 80-958 Gdańsk, Poland; wtrybowski@gmail.com; 3Department of Industrial and Food Microbiology, Faculty of Food Science, University of Warmia and Mazury in Olsztyn, Pl. Cieszyński 1, 10-726 Olsztyn, Poland; 4Department of Agroecosystems, Faculty of Environmental Management and Agriculture, University of Warmia and Mazury in Olsztyn, Pl. Łódzki 3, 10-719 Olsztyn, Poland; jozef.tyburski@uwm.edu.pl

**Keywords:** zearalenone, pre-pubertal gilts, GALT, oxidative stress, cytokine, metabolism

## Abstract

The aim of the present study was to determine the effect of zearalenone (ZEN), administered *per os* to gilts at doses equivalent to 50%, 100%, and 150% of no-observed-adverse-effect level (NOAEL) values for 14, 28, and 42 days during weaning, on changes in the parameters of the oxidoreductive balance, cytokine secretion, and basal metabolism in ileal Payer’s patches. Immunoenzymatic ELISA tests and biochemical methods were used to measure the concentrations of interleukin 1α, interleukin 1β, interleukin 12/23p40, interleukin 2, interferon γ, interleukin 4, interleukin 6, interleukin 8, tumor necrosis factor α, interleukin 10, transforming growth factor β, malondialdehyde, sulfhydryl groups, fructose, glucose, and proline, as well as the activity of peroxidase, superoxide dismutase and catalase. The study demonstrated that ZEN doses corresponding to 50%, 100%, and 150% of NOAEL values, i.e., 5 µg, 10 µg, and 15 µg ZEN/kg BW, respectively, have proinflammatory properties, exacerbate oxidative stress responses, and disrupt basal metabolism in ileal Payer’s patches in gilts.

## 1. Introduction

Zearalenone (ZEN) is regarded as one of the most frequently occurring mycotoxins in the world [1,2,3]. The structure of ZEN molecules enables the mycotoxin to bind to estrogen receptors and induce estrogenic effects. Cereals are the most frequent vectors of ZEN transmission [4], and according to the literature, 32% of 5010 mixed cereal samples analyzed in Europe were found to be ZEN positive [5]. Zearalenone is detected not only in fresh plants, but also in processed products, including pelleted feed, which can be attributed to the mycotoxin’s stability under exposure to increased temperature and pressure [6,7].

Pigs are regarded as a livestock that is very sensitive to ZEN. The above applies particularly to pre-pubertal gilts where ZEN induces symptoms of ovarian cysts, uterine edema, and early maturation of ovarian follicles [8,9]. In pre-pubertal gilts, exposure to ZEN causes edema and thickening of the vaginal wall and the lining of the vulva, increases uterus weight, and leads to ovarian atrophy and intensified proliferation of the epithelium of the uterine mucosa within 3 to 7 days [10]. In pregnant and lactating pigs, exposure to ZEN reduces the number of follicles in F1-generation piglets, which can lead to the premature recruitment of the oocyte pool in subsequent stages of development [11].

The gastrointestinal tract, a very important system in the body that digests and absorbs nutrients, is the first physiological barrier to ZEN [12,13]. Considerable differences in ZEN absorption have been observed in different segments of the small intestine, where 70%–80% of the mycotoxin was absorbed in the jejunum, and only 15%–30% in the ileum [14]. After crossing the intestinal barrier, ZEN enters the bloodstream and reaches the liver, where it is mainly biotransformed in hepatocytes in the presence of reducing factors such as NADPH, which leads to the transformation of ZEN to α-zearalenol (α-ZEL), β-zearalenol (β-ZEL), α-zearalanol (α-ZAL), and β-zearalanol (β-ZAL). Despite the fact that ZEN metabolites have different affinity for estrogen receptors ERα and ERβ [15], they disrupt endocrine functions in various animal species, both male and female [16]. Zearalenone’s estrogenic activity is 80 to 160 times weaker than that of 17-β-estradiol, but ZEN effectively competes with 17-β-estradiol, and the formed complex activates the transcription of estrogen-sensitive genes [16,17].

During continuous exposure to ZEN, the tissues of the porcine gastrointestinal tract remain under the constant influence of ZEN and its metabolites due to enterohepatic circulation of metabolites in pigs [18,19]. Zearalenone’s tropism for estrogen receptors plays a key role in the mycotoxin’s influence on the functions of gut-associated lymphoid tissue (GALT). Zearalenone exerts its effects mainly through ERα on T lymphocytes, natural killer (NK) cells, and macrophages, as well as ERβ that occur mainly on B lymphocytes and monocytes, and these processes are closely related to the mycotoxin’s effect on metabolic transformations in immunocompetent cells. Zearalenone also disrupts the oxidoreductive balance, which can lead to changes in cytokine expression in GALT.

Previous studies have demonstrated that progressing oxidative stress induced by ZEN in the gastrointestinal system increases the peroxidation of macromolecules, leading to changes to lipid membranes and proteins, and, consequently, a decrease in the barrier properties of the intestinal epithelium [20]. Reactive oxygen species (ROS) produced during lipid peroxidation promote the formation of 4-hydroxy-2-nonenal (4-HNE), malondialdehyde (MDA), propanal, and hexanal [21]. Malondialdehyde is the most mutagenic product, but 4-HNE is most toxic because it reacts with protein thiol and amino groups. Moreover, 4-HNE is metabolized to acetaldehyde, and together with MDA, it is bound to macromolecules via the amino groups of lysine, histidine, and arginine residues. The Schiff base is regrouped, which leads to the synthesis of lipid peroxidation end products. The same reactions take place in DNA, which induce the formation of protein/MDA/DNA intermolecular cross-links. These molecules induce inflammation, initiate complement activation, and play a role in the atherosclerotic process. By binding to mitochondrial proteins, MDA also modifies the activity of enzymes, membrane transport proteins, and cytoskeletal proteins.

GALT comprises different structures in the small intestine. It is composed of organized lymphoid follicles in the jejunum, whereas the ileum contains a continuous layer of aggregated lymphoid nodules (Peyer’s patches) that stretches from the ileum to the colon [22,23]. The functional significance of the organizational structure of Peyer’s patches remains unknown. It could be speculated that the massive accumulation of lymphoid tissue protects the boundary between the small intestine (jejunum and ileum) with moderate numbers of bacteria and the large intestine, which has an abundant microflora and contains potentially pathogenic microorganisms. The aim of the present experiment was to determine the effect of ZEN doses equivalent to 50% no-observed-adverse-effect level (NOAEL)—the no observable effect level (5 µg/kg BW), 100% NOAEL (10 µg/kg BW), and 150% NOAEL (15 µg/kg BW) values on changes in the oxidoreductive balance, cytokine secretion, and metabolic markers in ileal Peyer’s patches in pre-pubertal gilts during weaning.

## 2. Results

### 2.1. The Effect of Zearalenone on Cytokine Secretion

#### 2.1.1. Proinflammatory Cytokines

The concentration of IFN-γ in porcine ileal Peyer’s patches ranged from 9.52 to 52.45 pg/mg (Table 1). Significant differences were observed between groups and analytical dates. In groups ZEN II and ZEN III, the secretion of IFN-γ tended to increase with prolonged exposure to ZEN, and IFN-γ concentration increased by 29.73 and 11.03 ng/mg, respectively, relative to day 14 of the experiment. The noted increase was determined not only by the duration of exposure, but also by the administered ZEN dose. The concentration of IFN-γ peaked at 52.45 pg/mg on day 42 in response to a ZEN dose of 10 μg ZEN/kg BW (Table 1). On day 42, the concentration of IFN-γ decreased by 18.47 and 9.58 pg/mg in the control group and group ZEN I, respectively, relative to day 14.

In the remaining groups of proinflammatory cytokines, similar trends were noted in the secretion of interleukin (IL)-1α, IL-1β, and IL-2 (Table 2, Table 3 and Table 4). Significant differences were observed between experimental and control gilts and analytical dates. The greatest increase in IL-1α secretion was noted in response to ZEN doses of 10 μg/kg BW (ZEN II) and 15 μg/kg BW (ZEN III), in particular on days 14 and 42, and IL-1α levels increased by 322.70 and 351.90 pg/mg (ZEN II) and by 155.80 and 287.30 pg/mg (ZEN III), respectively, relative to the control group (Table 2).

Similar observations were made in the concentration of IL-1β, which increased by 53.6 and 94.59 pg/mg in group ZEN II, and by 41.10 and 88.19 pg/mg in group ZEN III on days 14 and 42, respectively (Table 3).

The secretion of IL-2 also increased in response to ZEN doses of 10 and 15 µg ZEN/kg BW. The highest increase in the concentration of IL-2 was noted on day 42 in group ZEN II where its content was 21.83 pg/mg higher than in the control gilts (Table 4).

The IL-6 secretion profile was highly similar in group ZEN I and in the control group. On day 42, the concentration of IL-6 increased significantly (*p* < 0.001) by 287.78 pg/mg in group ZEN II relative to the control group (Table 5).

In turn, IL-8 concentration tended to decrease throughout the experiment and was proportional to the administered ZEN dose (Table 6). The content of IL-8 decreased by 589.00 pg/mg in group ZEN I on day 42, by 906.00 pg/mg in group ZEN II on day 28, and by 929.20 pg/mg in group ZEN III on day 42 relative to the control group.

The concentration of IL-12/23p40, which is produced by, among others, macrophages, increased significantly (*p* < 0.001) after 42 days of administration to ZEN (Table 7), and it was 701.40 pg/mg higher than in the control group.

Similarly to IL-2, IL-1α and IL-1β, the secretion of tumor necrosis factor alpha (TNFα) was clearly affected by ZEN. The increasing trend in the concentration of TNFα was observed in groups ZEN II and ZEN III. The content of TNFα was highest in group ZEN III on day 42 when it exceeded TNFα levels in the control group by 149.84 pg/mg (*p* < 0.0001) (Table 8).

#### 2.1.2. Anti-Inflammatory and Regulatory Cytokines

The dynamics of IL-4 secretion was similar to the changes in the concentration of the proinflammatory IFN-γ. The concentration of IL-4 ranged from 277.40 to 2894.00 pg/mg. The highest increase in IL-4 content was noted in group ZEN II where the analyzed parameter was 2616.60 pg/mg higher on day 14 and 2094.30 pg/mg higher on day 42 than in the control group (Table 9).

The changes in the IL-10 profile were similar to those noted in the content of IL-4, IL-1α, IL-1β, IFN-γ, and IL-2. During the experiment, the concentration of IL-10 was lowest in group ZEN I. In turn, IL-10 levels increased in gilts administered ZEN doses of 10 and 15 pg/kg BW. The highest concentration of IL-10 relative to the control group was observed on day 42 in gilts from group ZEN II where it reached 62.61 pg/mg (Table 10).

Similarly to IL-8, the concentration of TGFβ decreased throughout the experiment, and significant differences were observed in groups ZEN II and III relative to the control group (Table 11).

### 2.2. The Effect of Zearalenone on Oxidative Stress Markers

Oxidative stress in the ileal Peyer’s patches of gilts was evaluated based on the activity of catalase (CAT), peroxidase (POD) and superoxide dismutase (SOD), MDA content, and the number of free metabolic sulfhydryl groups (-SH). The changes in the values of oxidative stress markers were similar to those noted in the concentrations of IL-1α, IL-1β, IFN-γ, IL-2, IL4, IL-10 and TNFα.

Peroxidase activity in the ileal Peyer’s patches of gilts ranged from 47.66 to 231.30 units/mg/min (Table 12).

Catalase activity ranged from 2.32 to 12.36 H_2_O_2_ [mM/mg/min] (Table 13). Significant differences were observed between experimental and control gilts and analytical dates. Catalase activity was strongly suppressed in gilts administered a ZEN dose of 10 µg ZEN/kg BW. The highest CAT activity was observed in group ZEN II.

The activity of SOD was highly similar to POD and CAT activity, and significant differences were observed between control and ZEN groups and analytical dates. The greatest increase in SOD activity was noted in group ZEN II where this parameter reached 17.94 H_2_O_2_ [mM/mg/min] on day 42 and was 7.62 H_2_O_2_ [mM/mg/min] higher than in the control gilts (Table 14).

The MDA profile in group ZEN I was highly similar to that noted in the control group. The concentration of MDA was highest on day 42 in groups ZEN II and ZEN III where it reached 1.43 and 1.39 pM/mg, respectively (Table 15).

During the experiment, the concentration of -SH groups decreased in gilts from group ZEN I. In turn, a very high increase in the concentration of -SH groups was noted in groups ZEN II and ZEN III. On day 42, this parameter reached 0.73 µM/mg in group ZEN II and 0.63 µM/mg in group ZEN III (Table 16).

### 2.3. The Effect of Zearalenone on Basal Metabolic Markers

Basal metabolism in ileal Peyer’s patches was evaluated based on the concentrations of glucose, fructose, and proline. In groups ZEN II and ZEN III, the values of metabolic markers were characterized by similar change trends to the values of oxidative stress markers (POD, CAT, SOD, MDA, -SH) and cytokine concentrations (IL-2, IFN-γ, IL-1α, IL-1β, IL4, IL-10, TNFα). Significant differences were observed mainly in groups ZEN II and ZEN III (Table 17, Table 18 and Table 19).

## 3. Discussion

In the European Union, the maximum limits for ZEN in livestock have been set at 0.25 mg ZEN/kg feed/day for adult pigs and 0.10 mg ZEN/kg feed/day for sows and piglets [24]. These limits have been introduced to protect pigs against the hyperestrogenic effects of ZEN. In morphometric analyses of porcine vulvae and uteri in animals exposed to ZEN, the NOAEL for ZEN was determined at 10.4 µg ZEN/kg BW/day for piglets and 40 µg ZEN/kg BW/day for mature female pigs [25]. When designing this study, the authors aimed to determine whether the established limits for protecting pigs against the estrogenic effects of ZEN are safe for GALT homeostasis. The theory proposed nearly 30 years ago [26] postulates that chemical processes involving oxidation-reduction reactions elicit oxidative stress and play a key role in the pathophysiology of inflammations [27]. This experiment was carry out on pre-pubertal gilts administered ZEN *per os* in doses equivalent to 50%, 100%, and 150% NOAEL values for 14, 28, and 42 days to evaluate the oxidoreductive balance, cytokine secretion, and basal metabolic markers in ileal Peyer’s patches. This research concept was formulated to address the general scarcity of the relevant data in the literature and to determine whether the elimination of ZEN’s estrogenic effects can also inhibit other adverse effects exerted by this mycotoxin.

Macrophages, mucosal cells, dendritic cells, NK cells, and neutrophils are responsible for innate immune defense in the digestive tract. The adaptive immune system is composed of T and B lymphocytes which, when activated, secrete cytokines and produce antibodies, respectively [28]. When the gastrointestinal mucosa is in a state of homeostasis, a strict balance is maintained between proinflammatory (IL-6, TNFα, IL-17, IL-1, IL-23, and IL-8) and anti-inflammatory cytokines (TGFβ, IL-5, IL-10, and IL-11), where neutrophils and other types of cells initiate inflammation by releasing proinflammatory interleukins to activate the adaptive immune response [29].

In the body, homeostasis is determined by the oxidant–antioxidant balance, which is most often disrupted by excessive production of ROS. Oxidation-reduction reactions are powerful chemical processes that involve oxidation and reduction of proteins and lipids, in particular unsaturated fatty acids, which generate superoxide radicals. The produced metabolites disrupt cell functions and regulatory mechanisms that are essential for maintaining homeostasis and a balance between health and disease [30].

Estradiol is a naturally occurring estrogen that delivers immunosuppressive effects when applied at high concentrations under experimental conditions. Previous research has demonstrated that low doses of mycoestrogens such as ZEN activated the porcine immune system [31,32]. The present study focused on evaluating the effect of low ZEN doses that do not exert estrogenic effects (5, 10 and 15 µg ZEN/kg BW) on GALT in the ileum’s walls of gilts during weaning. Zearalenone’s effects were evaluated based on cytokine secretion, the enzymatic and non-enzymatic markers of oxidative stress, and cellular metabolism in ileal Peyer’s patches.

The study demonstrated that each of the tested ZEN doses increased the secretion of proinflammatory cytokines (IFN-γ, IL-12/23p40, IL-1α, IL-1β, IL-2, IL-6, TNFα) in a different manner, which corroborates the findings of other authors [32,33]. In an in vitro study of the IPEC-2 cell line and mouse peritoneal macrophages, Fan et al. [34] noted an increase in the secretion of IL-1β and IL-18 under exposure to ZEN, and similar observations were made by Yousef et al. [35] in cultures of bovine oviductal epithelial cells. The present study also demonstrated that the concentrations of regulatory and anti-inflammatory cytokines (IL-4, IL-10) tended to increase, whereas TGFβ levels tended to decrease on successive days of exposure to ZEN. This is a very important observation, because TGFβ determines the integrity of the intestinal epithelium and is responsible for its repair as well as cell migration [36]. The binding of TGFβ signal molecules to the TGFβ IIR receptor may be disrupted in non-specific inflammatory bowel diseases (e.g., Crohn’s disease). In this experiment, each ZEN dose increased the concentrations of IL-2, IL-6, IL-12β, and IFN-γ in ileal Peyer’s patches. However, according to the literature, cytokine expression is determined not only by ZEN concentration, but also by the type of the exposed tissue or organ. Tarnau et al. [37] reported a decrease in the expression of IL-6, IL-8, IL-1β, TNFα, and IFN-γ genes in the liver and duodenum of gilts receiving a ZEN dose of 100 ppb. In contrast, the mRNA expression of the above cytokines increased in the pancreas, kidneys, and the colon, and IFN-γ expression increased in the liver. The real-time PCR method used in this study measures RNA levels; therefore, it does not have to be closely related to the direct concentration of cytokine protein. Despite the above, the expression of cytokine genes in the cited studies was similar to that noted in this experiment. Liu et al. [38] administered ZEN at 2.77 mg/kg feed to pregnant sows and reported an increase in the expression of genes encoding proinflammatory cytokines IL-6, IL-1α, IL-1β, and TNFα, but it did not observe changes in the expression of the IL-8 gene. The cited study revealed inflammation of the small intestine in pregnant sows as well as changes in newborn piglets where immune dysfunctions and increased expression of the genes encoding proinflammatory cytokines IL-6, IL-1α, IL-1β, and TNFα were observed in the small intestine.

Changes in the expression ratio of proinflammatory to anti-inflammatory cytokines can influence the permeability of the intestinal epithelium. Zearalenone can also affect the intestinal microbiome by decreasing the counts of selected bacterial strains, including Lactobacillus sp. [39]. Cytokines such as IL-12, IL-1α, IL-1β, TNFα, and IFN-γ are the key mediators of inflammation in GALT, whereas IL-10 and TGFβ should alleviate inflammatory states. According to some authors, ZEN induces intestinal inflammations via the NLRP3 inflammasome, which is composed of a NOD-like receptor and caspase-1 precursor. The NRLP inflammasome activated by ZEN is responsible for the maturation and secretion of proinflammatory cytokines IL-1β and IL-18, which play a role in the etiology of non-specific inflammatory bowel diseases. According to the literature, ZEN induces oxidative stress and increases ROS levels in the mitochondria, which also activates the NLRP inflammasome [34].

The intestinal epithelium significantly affects intestinal immunity because enterocytes form a protective barrier that shields the body from external influences. Enterocytes secrete retinoic acid metabolites, thymic stromal lymphopoietin (TSLP), and TGFβ, which induce immune tolerance to dietary antigens. Mycotoxins increase the permeability of the intestinal epithelium by influencing cell viability [40], and they decrease the proliferation of enterocytes and intestinal epithelial stem cells in intestinal crypts. Damage to the intestinal barrier allows the passage of intraluminal antigens, including bacterial antigens such as lipopolysaccharides (LPS), which stimulate macrophages and dendritic cells. Macrophages are polarized toward the M1 subpopulation, which increases the secretion of IL-12, IL-1β, and IL-1α [41]. A similar mechanism was observed in the current study, where the concentrations of IL-12/23p40, IL-1β, and IL-1α in ileal Peyer’s patches increased under exposure to ZEN doses of 10 and 15 µg ZEN/ kg BW. Elevated levels of IL-2 and IFN-γ are also indicative of the Th1/Tc1 response, which is induced by, among others, IL-12 secreted by macrophages. Macrophage metabolism also changes during inflammation because, according to the literature, M1 macrophages and dendritic cells activated by LPS undergo metabolic reprogramming [42]. This process is characterized by a decrease in the intensity of mitochondrial biochemical pathways, including the tricarboxylic acid cycle, respiratory chain, and fatty acid β-oxidation, as well as an increase in the activity of metabolic pathways in the cytoplasm, including glycolysis, the pentose phosphate pathway, and the synthesis of fatty acids and nucleic acids.

In the present study, glucose, fructose, and proline levels were elevated in gilts administered ZEN doses equivalent to 100% and 150% NOAEL values. According to the literature, mitogen-activated immune cells increase the demand for energy even 20-fold, which can be attributed to higher expression of the glucose transporter 1 (GLUT 1) gene [43]. The fact that glucose and fructose levels were higher than in the control gilts could be linked with the intensification of glycolysis and the pentose phosphate pathway, in particular in M1 macrophages and dendritic cells activated by LPS, whereas the increase in proline concentration in lymphoid tissues could be induced by activated metalloproteinases and the intensified synthesis of proline from glutamine.

Proline is a stress response marker in both plants and animals. In the current experiment, proline levels tended to increase in the GALT of gilts exposed to ZEN. According to the literature, inflammatory states lead to metabolic reprogramming as well as an increase in glucose uptake by cells and an increase in the expression of the GLUT 1 gene. An increase in glucose uptake, lactic acid production, and NADPH oxidase activity was reported in cultures of smooth muscle cells isolated from rat mesenteric veins and in human aortic smooth muscle cells treated with IL-1β [44]. Other authors demonstrated that the activation of T lymphocytes is determined by the amount of glucose transported into the cells, the GLUT 1 transporter, and the PI3K/Akt signaling pathway. Interestingly, a lower availability of glucose has been linked with a decrease in the IFN-γ secretion and proliferation of T cells. Despite the presence of other components, glucose is essential for the proliferation of T lymphocytes and IL-2 synthesis. Interleukin 10 exerts anti-inflammatory effects e.g., by blocking the expression of glycolytic enzymes and glucose transporters GLUT 1 and GLUT 4. In the present experiment, the progressive increase in IL-10 secretion was correlated with ZEN dose. However, other studies have demonstrated that the effects of IL-10 are reversible. Even low doses of estrogenic substances can stimulate IL-10 production by monocytes and macrophages [45] and by dorsal root ganglion cells in the nervous system [45].

Proline is also regarded as a marker of metabolic reprogramming. In pigs and humans, the small intestinal epithelium is the main site of proline synthesis from glutamate, but proline synthesized from arginine was not found in the intestinal epithelium of suckling piglets. Proline synthesis from glutamate is induced after weaning, and it is also linked with synthetase of pyrroline-5-carboxylic acid and proline oxidase. These enzymes are regulated by glucocorticoids [46], but also by lactic acid, which is present in quite high concentrations in the plasma of suckling animals. Research has also demonstrated that the synthesis of arginine and citrullin from proline in the intestinal epithelium decreases in environments with high concentrations of lactic acid [47]. In ileal Peyer’s patches, the synthesis of large quantities of lactic acid is linked with the metabolic reprogramming of M1 macrophages, high glycolytic activity, and high secretion of IL-12 and IL-1β [14]. M1 macrophages are produced after activation with bacterial products such as LPS during inflammations, including in the intestinal epithelium [47]. Lactic acid decreases the activity of proline oxidase, which catalyzes proline, an amino acid that participates in the biochemical transformation of arginine, a source of nitric oxide (NO). NO plays an important function in the regulation of the intestinal barrier function, and decreased proline metabolism can compromise intestinal integrity and lead to intestinal damage [47]. Interestingly, hypoargininemia was found to coexist with hyperargininemia and hyperlactacidemia in the plasma of patients with sepsis. During inflammations, proline can be released from the extracellular matrix, such as collagen, under the influence of metalloproteinases activated by stress responses and/or inflammation [48]. It should also be noted that proline and collagen synthesis can increase during inflammation, which can lead to tissue fibrosis [49]. More proline is synthesized from glutamine when the activity of the c-MYC transcription factor, also known as the oncogenic transcription factor, is intensified [50]. Proline contains an amine group with a ring structure, and unlike other amino acids, it does not undergo biochemical reactions, but is catabolized within the proline cycle [51]. This biochemical pathway occurs in the mitochondria, and its secondary effects include the production of superoxide anion radicals and an increase in the concentration of SOD. The proline cycle supports the generation of ATP when the activity of the respiratory chain decreases. The proline cycle also donates electrons to complexes III and IV of the respiratory chain, which is broken during inflammatory states caused by the metabolic reprogramming of cells [50,51]. Metabolic reprogramming also leads to oxidative stress that can be induced by ZEN.

An analysis of selected oxidative stress parameters revealed a correlation between immunological, metabolic, and oxidative stress parameters. Superoxide dismutase activity was proportional to proline content as well as CAT and POD activity. The activity of CAT, POD, and SOD increases at the beginning of inflammation, and it appears to be a defense response [52,53]. Oxidative stress is also linked with the activity of the inflammasome, IL-1α, and IL-1β, and it increases the permeability of the intestinal barrier, which affects the intestinal epithelium [54]. In reality, these processes are linked by biochemical pathways, and they are also dependent on the ZEN dose. The concentration of proinflammatory cytokines (IL-1β, TNFα, IFNγ) in the liver and spleen decreased in gilts fed diets containing 316 µg ZEN/kg feed [55]. According to the literature, the expression of CAT and glutathione peroxidase genes was elevated in oxidative stress [55]. In the current experiment, the values of metabolic, oxidative stress, and immunological parameters peaked under exposure to a ZEN dose of 10 µg ZEN/kg feed. However, in piglets administered 15 µg ZEN/kg feed, these parameters decreased or showed a decreasing trend relative to the previous dose, but, interestingly, their values were still higher than in the control gilts. In other studies, the values of immunological and oxidative stress parameters in the kidneys increased in pigs administered ZEN doses of 0.0003, 0.048, 0.098, and 0.146 g ZEN/kg feed.

In the present study, the concentrations of metabolic sulfhydryl groups (-SH) increased in ileal Peyer’s patches. Metabolic -SH groups are linked with the pentose phosphate pathway and glycolysis, which are elements of metabolic reprogramming in cells during inflammation [39,56]. Cysteine residues in proteins and glutathione (GSH) are the main donors of -SH groups in cells. The reduced form of glutathione (GSSG) is synthesized by glutathione reductase. The activity of this enzyme is linked with intracellular redox states and reduced forms of nicotinamide adenine coenzymes NADH and NADPH [22]. The supply of reduced coenzymes is strictly correlated with the activity of the glycolysis pathway and the pentose phosphate pathway. Moreover, GSH also stimulates glycolysis and glutaminolysis, which are characteristic of metabolic reprogramming of active T lymphocytes via the myc transcription factor [39]. Catalase activity can be closely linked with the level of NADPH because this coenzyme is molecularly bound to CAT. For example, human CAT contains four molecules of tightly bound NADPH [39]. At the same time, metabolic changes decrease oxidative phosphorylation, which disrupts the respiratory chain and leads to the production of ROS and hydrogen peroxide. The above intensifies the peroxidation of cell membrane lipids, increases MDA concentration, and enhances the activity of cellular scavenger enzymes: CAT and POD [57]. Malondialdehyde is a reactive compound that is formed during reactions involving ROS (mainly superoxide anion radicals and hydrogen peroxide) and unsaturated fatty acids which are found in plasma membranes and endoplasmic reticulum in mitochondrial, microsomal, and nuclear membranes. Reactive oxygen species are produced during inflammatory states in the mitochondria, where metabolic reprogramming contributes to alterations in the Krebs cycle. The above leads to the accumulation of succinic acid, a decrease in the concentrations of reduced forms of flavin adenine dinucleotide (FAD) and nicotinamide adenine dinucleotide (NAD), and uncoupling of the respiratory chain. Reactive oxygen species are most dynamically formed in complexes I and III of the electron transport chain [50].

## 4. Conclusions

To sum up, the close links between metabolic, oxidative stress, and immunological parameters during inflammatory states testify to the presence of a single process that begins with biochemical changes in metabolism in response to stress factors. The results of the present study indicate that ZEN doses are equivalent to 50%, 100%, and 150% NOAEL values, which induce such changes in ileal Peyer’s patches. It could be postulated that cell responses to low ZEN doses initiate reactions that lead to inflammation. The proinflammatory properties of ZEN and intensified oxidative stress can impair the function of the intestinal epithelium, as demonstrated by oxidative stress markers such as the biochemical changes associated with the metabolism of sugars (enhanced glycolysis) and amino acids (proline).

## 5. Materials and Methods 

### 5.1. Animals and Diet

Experiments were carried out in compliance with Polish regulations setting forth the terms and conditions of animal experimentation (Opinions No. 12/2016 and 45/2016/DLZ of the Local Ethics Committee for Animal Experimentation of 27 April 2016 and 30 November 2016).

A total of 60 gilts (14.5 ± 2 kg BW) were used in the experiment. The animals were acclimated in the experimental facilities of the Faculty of Veterinary Medicine of the University of Warmia and Mazury in Olsztyn (Poland) for one week before the experiment. Gilts were penned in groups with ad libitum access to feed and water throughout the experiment. The animals were randomly divided into three experimental groups (ZEN I, ZEN II, and ZEN III; n = 15) and a control group (Control; n = 15). The experimental gilts were orally administered ZEN (Z2125-25MG, Sigma-Aldrich, USA) at a dose of 5 μg ZEN/kg BW (group ZEN I), 10 μg ZEN/kg BW (group ZEN II), and 15 μg ZEN/kg BW (group ZEN III). The appropriate analytical doses of mycotoxins were dissolved in 96% ethanol, combined with the feed carrier and placed in gel capsules (Polskie Odczynniki SA, Gliwice, Poland). Before administration, open capsules were stored to evaporate ethanol. The capsules were administered daily before morning feeding. The gilts were weighed every week to adjust the ZEN dose to the body weight of each pig.

All animals were fed pelleted feed supplied by the same producer. The gilts were fed *ad libitum* at 08:00 and 17:00 during the entire experiment. The composition of pelleted feed is presented in Table 20 [58].

### 5.2. Tissue Sampling and Preparation for Analyses

The experimental material comprised segments of the ileum collected from gilts after 14, 28, and 42 days of exposure to ZEN. On each sampling date, five gilts selected randomly from each group were premedicated with azaperone in dose 4 mg/kg BW, im (Stresnil, Jansen Pharmaceutica NV, Belgium) and euthanized (sodium pentobarbital, 0.6 ml/kg BW, iv) after 15 min (Morbital, Biowet Puławy, Poland). Segments of the ileum (with a length of 5 cm each, sampled from the same site in every animal) located 2 cm away from the ileocecal valve were collected immediately after euthanasia. The samples were stored at a temperature of −80 °C until cytokine analysis.

Samples of 1 g of minced ileum were processed with 2.5 ml of the extraction dilution (137 mM NaCl, 2.7 mM KCl, 8.1 mM Na_2_HPO_4_, 1.5 mM KH_2_PO_4_), 0.5% sodium citrate (Avantor, Poland), 0.05% Tween 20 (Serva, Germany), protease inhibitors (Roche, Germany)] in a homogenizer (Omni-TipsTM Disposable, Omni International, Kennesaw, GA, USA). The homogenate was centrifuged (8600 g for 1 hour) in an Eppendorf 5804R centrifuge, and supernatant samples were stored at −80 °C until analysis.

### 5.3. Immunoenzymatic Determination of Cytokines and Oxidative Stress Markers

To determine the concentrations of cytokines in porcine tissues, commercial ELISA kits were used (Table 21). The tissues were homogenized in radioimmunoprecipitation assay buffer (RIPA) buffer in 4 °C and were centrifuged (30,000× *g* for 1 h). After centrifugation, the obtained tissue supernatants were aliquoted and stored at −80 °C. The supernatants were used for cytokines measurements. The ELISA test plate was measured with the TECAN Infinite M200PRO (Austria) plate reader at a wavelength of λ = 492 nm. The concentrations of cytokines in the tissues were measured by the bicinchoninic acid (BCA) method (Pierce BCA Protein Assay Kit, Thermo Scientific, Rockford, IL, USA) and expressed per milligram of protein.

### 5.4. Determination of Malondialdehyde Levels (Thiobarbituric Acid Assay)

The level of malondialdehyde (MDA) was measured according to the method described by Weitner et al. [59] with small modifications. Briefly, the tissue supernatant with buthylohydroxytoluene (BHT, Sigma Aldrich, Saint Louis, MO, USA) was deproteinized with 20% TCA (trichloroacetic acid, Avantor, Poland) and centrifuged for 1 h; 100 μl of the supernatant was then combined with thiobarbituric acid (TBA, Sigma Aldrich, Saint Louis, MO, USA) and incubated (1 h at 95°C). The level of MDA was read from a calibration curve (TBA, MDA Standard, Cayman, Ann Arbor, MI, USA). Absorbance was read in a Perkin Elmer spectrophotometer Lambda 25 at a wavelength of λ= 520 nm (Biocompare, Baltimore, MD, USA). The level of MDA was expressed in picomoles per milligram of whole protein in the tissue supernatant.

### 5.5. Determination of Sulfhydryl Groups (–SH)

Thiol groups were measured according to the modified Ellman method. Briefly, 1.0 ml of 40 mM Ellman’s reagent (5,5’-dithiobis-(2-nitrobenzoic acid) (DTNB) solution (Serva, Heidelberg, Germany) was added to the sample (86 mM Tris (Sigma Aldrich, Saint Louis, MO, USA), 90 mM glycine (Avantor, Gliwice, Poland), 4 mM ethylenediaminetetraacetic acid (EDTA) (Avantor, Gliwice, Poland), 8 M urea (Sigma Aldrich, Saint Louis, MO, USA), 0.5% sodium dodecyl sulfate (SDS, Serva, Heidelberg, Germany), 0.2 M Tris-HCl (Sigma Aldrich, Saint Louis, MO, USA)) with pH 8.0. Next, 200 μl of each sample was added to 1.0 ml of DTNB. The samples were incubated at room temperature for 30 min. Cysteine was used (Avantor, Gliwice, Poland) as a standard, and absorbance was measured by a Perkin Elmer spectrophotometer Lambda 25 at a wavelength of λ= 412 nm (Biocompare, Baltimore, MD, USA). The concentration of -SH groups was measured from a calibration curve based on cysteine solution in PBS. The concentration of -SH groups was expressed in micromoles per milligram of whole protein in the ileal supernatant.

### 5.6. Determination of Fructose and Glucose Concentrations

Fructose level was determined in the ileum by the method described by Messineo and Musarra [60] with some modifications. This method is specific for fructose and similar to sucrose and inulin measurements method without interference from glucose (aldohexoses), aldopentose, and ketopentose. Glucose concentration was evaluated by Trinder’s glucose oxidase method modified by Lott and Turner [61], with further modifications. The measurements were conducted spectrophotometrically with glucose oxidase reagent (G7521, Pointe Scientific, Canton, MI, USA) with the appropriate modifications. Fructose and glucose concentrations were expressed in micrograms per milligram of whole protein in the ileal supernatant.

### 5.7. Determination of Proline Concentration

Proline concentration in ileal Peyer’s patches was determined by the modified method for the determination of proline levels in plants [62]. Tissue homogenates in the amount of 500 mL were placed in glass test tubes, and 1.0 mL of 2.5% ninhydrin solution, 1.0 mL of 3% sulfosalicylic acid, and 1.0 mL of glacial acetic acid were added. The samples were incubated (boiling water bath for 15 min.). The glass test tubes were cooled under running water, and 2 mL of toluene was added. The tubes were shaken for 15 min and left to stand until the separation of mixture components. The toluene layer was measured spectrophotometrically at a wavelength of 520 nm. Proline concentration was read from the standard curve, and points on the curve were determined based on the sample preparation method.

### 5.8. Statistical Analysis

The results were processed in Excel (Microsoft, Redmond, WA, USA) and GraphPad Prism 6 (GraphPad Software, San Diego, CA, USA) applications. Mean values and standard error of the mean (SEM) were determined for all groups. Population distributions were evaluated by the Shapiro–Wilk normality test. The results were processed by two-way ANOVA with post hoc Tukey’s multiple comparison test. The results were regarded as statistically significant at *p* < 0.05.

## Figures and Tables

**Table 1 toxins-12-00350-t001:** Changes in IFN-γ content in the ileum. Experimental group and zearalenone (ZEN) concentration: Control Group, ZEN I—5 μg/kg BW, ZEN II—10 μg/kg BW, ZEN III—15 μg/kg BW.

Group	Experimental Day					
	14		28		42	
	mean	SEM	mean	SEM	mean	SEM
Control	35.98	9.66	17.33	2.84	17.51 ^A^	2.49
ZEN I	19.10	7.88	10.60	1.69	9.52 ^B^	0.70
ZEN II	22.72	3.31	16.55	2.59	52.45 ^C^	10.37
ZEN III	23.66	5.39	28.61	7.28	34.69	3.17

The results are expressed as means ± SEM (standard error of the mean). Statistically significant differences: 42 day, ^A^ Control Group vs. ZEN II, *p* < 0.01; ^B^ ZEN I vs. ZEN II, *p* < 0.001; ^C^ ZEN I vs. ZEN III, *p* < 0.05.

**Table 2 toxins-12-00350-t002:** Changes in IL-1α content in the ileum. Experimental group and ZEN concentration: Control Group, ZEN I—5 μg/kg BW, ZEN II—10 μg/kg BW, ZEN III—15 μg/kg BW.

Group	Experimental day					
	14		28		42	
	mean	SEM	mean	SEM	mean	SEM
Control	307.30 ^A^	12.12	352.70	30.74	281.00 ^C,D^	16.87
ZEN I	283.10 ^B^	32.76	267.50	20.80	161.20 ^E,F^	8.16
ZEN II	630.00	72.20	390.70	33.47	632.90	107.90
ZEN III	463.10	81.18	417.90	77.07	568.30	55.67

The results are expressed as means ± SEM (standard error of the mean). Statistically significant differences: 14 day, ^A^ Control Group vs. ZEN II, *p* < 0.001; ^B^ ZEN I vs. ZEN II, *p* < 0.001; 42 day, ^C^ Control Group vs. ZEN II, *p* < 0.0001, ^D^ Control Group vs. ZEN III, *p* < 0.01, ^E^ ZEN I vs. ZEN II, *p* < 0.0001, ^F^ ZEN I vs. ZEN III, *p* < 0.0001.

**Table 3 toxins-12-00350-t003:** Changes in IL-1β content in the ileum. Experimental group and ZEN concentration: Control Group, ZEN I—5 μg/kg BW, ZEN II—10 μg/kg BW, ZEN III—15 μg/kg BW.

Group	Experimental Day					
	14		28		42	
	mean	SEM	mean	SEM	mean	SEM
Control	113.70	9.32	114.60	9.12	99.01 ^B,C^	8.78
ZEN I	93.11 ^A^	10.43	91.40	9.74	75.80 ^D,E^	9.87
ZEN II	167.30	22.20	86.21	9.89	193.60	37.13
ZEN III	154.80	22.13	103.60	27.92	187.20	21.79

The results are expressed as means ± SEM (standard error of the mean). Statistically significant differences: 14 day, ^A^ ZEN I vs. ZEN II, *p* < 0.05; 42 days, ^B^ Control Group vs. ZEN II, *p* < 0.01, ^C^ Control Group vs. ZEN III, *p* < 0.01, ^D^ ZEN I vs. ZEN II, *p* < 0.001, ^E^ ZEN I vs. ZEN III, *p* < 0.001.

**Table 4 toxins-12-00350-t004:** Changes in IL-2 content in the ileum. Experimental group and ZEN concentration: Control Group, ZEN I—5 μg/kg BW, ZEN II—10 μg/kg BW, ZEN III—15 μg/kg BW.

Group	Experimental Day					
	14		28		42	
	mean	SEM	mean	SEM	mean	SEM
Control	6.39 ^A,B^	1.21	11.93	1.78	10,10 ^E,F^	1.29
ZEN I	6.56 ^C,D^	0.88	8.97	1.30	7.43 ^G,H^	1.46
ZEN II	28.68	2.67	22.52	2.90	55.04 ^I^	13.55
ZEN III	24.81	4.65	22.03	4.19	31.93	1.28

The results are expressed as means ± SEM (standard error of the mean). Statistically significant differences: 14 day, ^A^ Control Group vs. ZEN II, *p* < 0.01, ^B^ Control Group vs. ZEN III, *p* < 0.05, ^C^ ZEN I vs. ZEN II, *p* < 0.01, ^D^ ZEN I vs. ZEN III, *p* < 0.05; 42 day, ^E^ Control Group vs. ZEN II, *p* < 0.0001, ^F^ Control Group vs. ZEN III, *p* < 0.01, ^G^ ZEN I vs. ZEN II, *p* < 0.0001, ^H^ ZEN I vs. ZEN III, *p* < 0.01, ^I^ ZEN II vs. ZEN III, *p* < 0.01.

**Table 5 toxins-12-00350-t005:** Changes in IL-6 content in the ileum. Experimental group and ZEN concentration: Control Group, ZEN I—5 μg/kg BW, ZEN II—10 μg/kg BW, ZEN III—15 μg/kg BW.

Group	Experimental Day					
	14		28		42	
	mean	SEM	mean	SEM	mean	SEM
Control	45.44	4.86	67.76	12.72	47.72 ^A^	5.39
ZEN I	41.66	5.50	54.35	6.11	47.15 ^B^	8.31
ZEN II	141.80	13.12	160.20	21.34	345.50 ^C^	175.80
ZEN III	88.69	12.10	103.20	17.43	126.90	6.81

The results are expressed as means ± SEM (standard error of the mean). Statistically significant differences: 42 day, ^A^ Control Group vs. ZEN II, *p* < 0.001, ^B^ ZEN I vs. ZEN II, *p* < 0.001, ^C^ ZEN II vs. ZEN III, *p* < 0.05.

**Table 6 toxins-12-00350-t006:** Changes in IL-8 content in the ileum. Experimental group and ZEN concentration: Control Group, ZEN I—5 μg/kg BW, ZEN II—10 μg/kg BW, ZEN III—15 μg/kg BW.

Group	Experimental Day					
	14		28		42	
	mean	SEM	mean	SEM	mean	SEM
Control	1390.00	79.32	1916.00 ^A,B^	266.60	1619.00	177.20
ZEN I	1400.00	141.90	1840.00 ^C,D^	147.60	1030.00 ^E^	89.83
ZEN II	1502.00	150.60	1010.00	114.20	1700.00	321.90
ZEN III	1203.00	165.30	986.80	163.30	1267.00	64.91

The results are expressed as means ± SEM (standard error of the mean). Statistically significant differences: 28 day, ^A^ Control Group vs. ZEN II, *p* < 0.01, ^B^ Control Group vs. ZEN III, *p* < 0.01, ^C^ ZEN I vs. ZEN II, *p* < 0.01, ^D^ ZEN I vs. ZEN III, *p* < 0.01; 42 day, ^E^ ZEN I vs. ZEN II, *p* < 0.05.

**Table 7 toxins-12-00350-t007:** Changes in IL-12/23p40 content in the ileum. Experimental group and ZEN concentration: Control Group, ZEN I—5 μg/kg BW, ZEN II—10 μg/kg BW, ZEN III—15 μg/kg BW.

Group	Experimental Day					
	14		28		42	
	mean	SEM	mean	SEM	mean	SEM
Control	873.60	76.30	918.90	67.70	827.60 ^A^	38.05
ZEN I	653.70	117.30	823.80	87.98	539.50 ^B^	42.28
ZEN II	762.90	89.15	652.00	68.79	1529.00 ^C^	323.40
ZEN III	753.50	36.76	590.90	121.10	763.80	99.28

The results are expressed as means ± SEM (standard error of the mean). Statistically significant differences: 42 day, ^A^ Control Group vs. ZEN II, *p* < 0.001, ^B^ ZEN I vs. ZEN II, *p* < 0.0001, ^C^ ZEN II vs. ZEN III, *p* < 0.001.

**Table 8 toxins-12-00350-t008:** Changes in TNFα content in the ileum. Experimental group and ZEN concentration: Control Group, ZEN I—5 μg/kg BW, ZEN II—10 μg/kg BW, ZEN III—15 μg/kg BW.

Group	Experimental Day					
	14		28		42	
	mean	SEM	mean	SEM	mean	SEM
Control	38.45 ^A^	6.52	100.30	15.27	46.16 ^B,C^	6.79
ZEN I	100.90	32.73	98.03	21.61	49.26 ^D,E^	9.38
ZEN II	132.90	17.87	111.60	9.35	196.00	41.90
ZEN III	97.99	10.89	135.50	20.92	125.20	7.20

The results are expressed as means ± SEM (standard error of the mean). Statistically significant differences: 14 day, ^A^ Control Group vs. ZEN II, *p* < 0.01; 42 days, ^B^ Control Group vs. ZEN II, *p* < 0.0001, ^C^ Control Group vs. ZEN III, *p* < 0.05, ^D^ ZEN I vs. ZEN II, *p* < 0.0001, ^E^ ZEN I vs. ZEN III, *p* < 0.05.

**Table 9 toxins-12-00350-t009:** Changes in IL-4 content in the ileum. Experimental group and ZEN concentration: Control Group, ZEN I—5 μg/kg BW, ZEN II—10 μg/kg BW, ZEN III—15 μg/kg BW.

Group	Experimental Day					
	14		28		42	
	mean	SEM	mean	SEM	mean	SEM
Control	277.40 ^A^	56.61	572.30	130.40	470.70 ^D^	91.95
ZEN I	356.80 ^B^	56.00	463.80	77.36	330.90 ^E^	56.96
ZEN II	2894.00 ^C^	955.90	1352.00	318.90	2565.00 ^F^	680.60
ZEN III	589.80	91.53	994.20	260.70	908.50	93.04

The results are expressed as means ± SEM (standard error of the mean). Statistically significant differences: 14 day, ^A^ Control Group vs. ZEN II, *p* < 0.001, ^B^ ZEN I vs. ZEN II, *p* < 0.001, ^C^ ZEN II vs. ZEN III, *p* < 0.01; 42 day, ^D^ Control Group vs. ZEN II, *p* < 0.01, ^E^ ZEN I vs. ZEN II, *p* < 0.01, ^F^ ZEN II vs. ZEN III, *p* < 0.05.

**Table 10 toxins-12-00350-t010:** Changes in IL-10 content in the ileum. Experimental group and ZEN concentration: Control Group, ZEN I—5 μg/kg BW, ZEN II—10 μg/kg BW, ZEN III—15 μg/kg BW.

Group	Experimental Day					
	14		28		42	
	mean	SEM	mean	SEM	mean	SEM
Control	37.40	6.45	57.49 ^B^	8.16	35.35 ^C^	4.89
ZEN I	23.69 ^A^	6.44	25.87	2.70	14.46 ^D,E^	1.47
ZEN II	52.33	6.76	39.31	2.41	77.06	15.46
ZEN III	42.51	5.36	44.76	7.42	56.36	3.66

The results are expressed as means ± SEM (standard error of the mean). Statistically significant differences: 14 day, ^A^ ZEN I vs. ZEN II, *p* < 0.05; 28 day, ^B^ Control Group vs. ZEN I, *p* < 0.01; 42 day, ^C^ Control Group vs. ZEN II, *p* < 0.001, ^D^ ZEN I vs. ZEN II, *p* < 0.0001, ^E^ ZEN I vs. ZEN III, *p* < 0.001.

**Table 11 toxins-12-00350-t011:** Changes in TGFβ content in the ileum. Experimental group and ZEN concentration: Control Group, ZEN I—5 μg/kg BW, ZEN II—10 μg/kg BW, ZEN III—15 μg/kg BW.

Group	Experimental Day					
	14		28		42	
	mean	SEM	mean	SEM	mean	SEM
Control	215.20 ^A^	19.46	263.30 ^B,C^	19.46	134.50	15.71
ZEN I	143.40	23.98	212.20 ^D,E^	28.75	160.20	13.42
ZEN II	125.60	15.56	116.60	14.16	129.90	8.25
ZEN III	193.80	28.69	126.50	7.21	170.20	39.46

The results are expressed as means ± SEM (standard error of the mean). Statistically significant differences: 14 day, ^A^ Control Group vs. ZEN III, *p* < 0.05; 28 days, ^B^ Control Group vs. ZEN II, *p* < 0.001, ^C^ Control Group vs. ZEN III, *p* < 0.0001, ^D^ ZEN I vs. ZEN II, *p* < 0.05, ^E^ ZEN I vs. ZEN III, *p* < 0.05.

**Table 12 toxins-12-00350-t012:** Changes in peroxidase activity in the ileum. Experimental group and ZEN concentration: Control Group, ZEN I—5 μg/kg BW, ZEN II—10 μg/kg BW, ZEN III—15 μg/kg BW.

Group	Experimental Day					
	14		28		42	
	mean	SEM	mean	SEM	mean	SEM
Control	74.88 ^A,B^	4.56	74.63 ^E,F^	4.85	87.04 ^I,J^	6.63
ZEN I	47.66 ^C,D^	5.42	79.13 ^G,H^	3.09	73.62 ^K,L^	5.64
ZEN II	184.40	19.23	156.90	13.99	231.30	19.70
ZEN III	157.90	16.06	179.20	29.34	214.80	16.03

The results are expressed as means ± SEM (standard error of the mean). Statistically significant differences: 14 day, ^A^ Control Group vs. ZEN II, *p* < 0.0001, ^B^ Control Group vs. ZEN III, *p* < 0.001, ^C^ ZEN I vs. ZEN II, *p* < 0.0001, ^D^ ZEN I vs. ZEN III, *p* < 0.0001; 28 day, ^E^ Control Group vs. ZEN II, *p* < 0.001, ^F^ Control Group vs. ZEN III, *p* < 0.0001, ^G^ ZEN I vs. ZEN II, *p* < 0.01, ^H^ ZEN I vs. ZEN III, *p* < 0.0001; 42 day, ^I^ Control Group vs. ZEN II, *p* < 0.0001, ^J^ Control Group vs. ZEN III, *p* < 0.0001, ^K^ ZEN I vs. ZEN II, *p* < 0.0001, ^L^ ZEN I vs. ZEN III, *p* < 0.0001.

**Table 13 toxins-12-00350-t013:** Changes in catalase activity in the ileum. Experimental group and ZEN concentration: Control Group, ZEN I—5 μg/kg BW, ZEN II—10 μg/kg BW, ZEN III—15 μg/kg BW.

Group	Experimental Day					
	14		28		42	
	mean	SEM	mean	SEM	mean	SEM
Control	6.67 ^A^	0.35	6.49 ^E^	0.40	3.92 ^H,I^	0.25
ZEN I	4.24 ^B,C^	0.62	3.58 ^F,G^	0.38	2.32 ^J,K^	0.14
ZEN II	12.23 ^D^	1.12	9.20	0.81	12.36 ^L^	1.50
ZEN III	7.65	0.61	7.83	1.09	8.73	0.52

The results are expressed as means ± SEM (standard error of the mean). Statistically significant differences: 14 day, ^A^ Control Group vs. ZEN II, *p* < 0.0001, ^B^ ZEN I vs. ZEN II, *p* < 0.0001, ^C^ ZEN I vs. ZEN III, *p* < 0.01, ^D^ ZEN II vs. ZEN III, *p* < 0.001; 28 day, ^E^ Control Group vs. ZEN I, *p* < 0.05, ^F^ ZEN I vs. ZEN II, *p* < 0.0001, ^G^ ZEN I vs. ZEN III, *p* < 0.001; 42 day, ^H^ Control Group vs. ZEN II, *p* < 0.0001, ^I^ Control Group vs. ZEN III, *p* < 0.0001, ^J^ ZEN I vs. ZEN II, *p* < 0.0001, ^K^ ZEN I vs. ZEN III, *p* < 0.01.

**Table 14 toxins-12-00350-t014:** Changes in the activity of superoxide dismutase in the ileum. Experimental group and ZEN concentration: Control Group, ZEN I—5 μg/kg BW, ZEN II—10 μg/kg BW, ZEN III—15 μg/kg BW.

Group	Experimental Day					
	14		28		42	
	mean	SEM	mean	SEM	mean	SEM
Control	11.35 ^A^	0.65	11.63	0.78	10.32 ^E,F^	0.32
ZEN I	8.14 ^B^	0.91	7.31 ^C,D^	0.73	1.14 ^G,H^	0.10
ZEN II	17.94	1.55	14.12	1.28	18.63	2.75
ZEN III	13.57	1.33	14.62	2.54	15.80	0.36

The results are expressed as means ± SEM (standard error of the mean). Statistically significant differences: 14 day, ^A^ Control Group vs. ZEN II, *p* < 0.05, ^B^ ZEN I vs. ZEN II, *p* < 0.001; 28 day, ^C^ ZEN I vs. ZEN II, *p* < 0.05, ^D^ ZEN I vs. ZEN III, *p* < 0.01; 42 day, ^E^ Control Group vs. ZEN I, *p* < 0.001, ^F^ Control Group vs. ZEN II, *p* < 0.01, ^G^ ZEN I vs. ZEN II, *p* < 0.0001, ^H^ ZEN I vs. ZEN III, *p* < 0.0001.

**Table 15 toxins-12-00350-t015:** Changes in the concentration of malondialdehyde in the ileum. Experimental group and ZEN concentration: Control Group, ZEN I—5 μg/kg BW, ZEN II—10 μg/kg BW, ZEN III—15 μg/kg BW.

Group	Experimental Day					
	14		28		42	
	mean	SEM	mean	SEM	mean	SEM
Control	0.28 ^A^	0.08	0.67	0.34	0.41 ^C,D^	0.11
ZEN I	0.25 ^B^	0.12	0.49	0.18	0.32 ^E,F^	0.20
ZEN II	1.26	0.12	0.65	0.06	1.43	0.50
ZEN III	0.54	0.07	0.83	0.15	1.39	0.52

The results are expressed as means ± SEM (standard error of the mean). Statistically significant differences: 14 day, ^A^ Control Group vs. ZEN II, *p* < 0.05, ^B^ ZEN I vs. ZEN II, *p* < 0.05; 42 day, ^C^ Control Group vs. ZEN II, *p* < 0.05, ^D^ Control Group vs. ZEN III, *p* < 0.05, ^E^ ZEN I vs. ZEN II, *p* < 0.05, ^F^ ZEN I vs. ZEN III, *p* < 0.05.

**Table 16 toxins-12-00350-t016:** Changes in the content of -SH groups in the ileum. Experimental group and ZEN concentration: Control Group, ZEN I—5 μg/kg BW, ZEN II—10 μg/kg BW, ZEN III—15 μg/kg BW.

Group	Experimental Day					
	14		28		42	
	mean	SEM	mean	SEM	mean	SEM
Control	0.27 ^A, B^	0.01	0.30 ^E, F^	0.02	0.27 ^I, J^	0.01
ZEN I	0.21 ^C, D^	0.02	0.26 ^G, H^	0.02	0.17 ^K, L^	0.01
ZEN II	0.63	0.04	0.53	0.04	0.77	0.09
ZEN III	0.51	0.04	0.54	0.07	0.63	0.02

The results are expressed as means ± SEM (standard error of the mean). Statistically significant differences: 14 day, ^A^ Control Group vs. ZEN II, *p* < 0.0001, ^B^ Control Group vs. ZEN III, *p* < 0.001, ^C^ ZEN I vs. ZEN II, *p* < 0.0001, ^D^ ZEN I vs. ZEN III, *p* < 0.0001; 28 day, ^E^ Control Group vs. ZEN II, *p* < 0.001, ^F^ Control Group vs. ZEN III, *p* < 0.001, ^G^ ZEN I vs. ZEN II, *p* < 0.0001, ^H^ ZEN I vs. ZEN III, *p* < 0.0001, 42 day, ^I^ control group vs. ZEN II, *p* < 0.0001, ^J^ Control Group vs. ZEN III, *p* < 0.0001, ^K^ ZEN I vs. ZEN II, *p* < 0.0001, ^L^ ZEN I vs. ZEN III, *p* < 0.0001.

**Table 17 toxins-12-00350-t017:** Changes in glucose content in the ileum. Experimental group and ZEN concentration: Control Group, ZEN I—5 μg/kg BW, ZEN II—10 μg/kg BW, ZEN III—15 μg/kg BW.

Group	Experimental Day					
	14		28		42	
	mean	SEM	mean	SEM	mean	SEM
Control	5.60 ^A,B^	1.68	11.79 ^D^	2.71	20.59 ^E,F^	1.98
ZEN I	23.20 ^C^	2.45	15.67	2.41	16.39 ^G,H^	1.73
ZEN II	44.93	8.86	40.80	4.36	80.79 ^I^	20.29
ZEN III	62.39	6.92	29.37	3.69	51.74	11.37

The results are expressed as means ± SEM (standard error of the mean). Statistically significant differences: 14 day, ^A^ Control Group vs. ZEN II, *p* < 0.01, ^B^ Control Group vs. ZEN III, *p* < 0.0001, ^C^ ZEN I vs. ZEN III, *p* < 0.01; 28 day, ^D^ Control Group vs. ZEN II, *p* < 0.05; 42 day, ^E^ Control Group vs. ZEN II, *p* < 0.0001, ^F^ Control Group vs. ZEN III, *p* < 0.05, ^G^ ZEN I vs. ZEN II, *p* < 0.0001, ^H^ ZEN I vs. ZEN III, *p* < 0.01, ^I^ ZEN II vs. ZEN III, *p* < 0.05.

**Table 18 toxins-12-00350-t018:** Changes in fructose content in the ileum. Experimental group and ZEN concentration: Control Group, ZEN I—5 μg/kg BW, ZEN II—10 μg/kg BW, ZEN III—15 μg/kg BW.

Group	Experimental Day					
	14		28		42	
	mean	SEM	mean	SEM	mean	SEM
Control	5.24 ^A,B^	0.26	4.94 ^F^	0.32	5.18 ^H,I^	0.25
ZEN I	5.78 ^C,D^	0.69	4.98 ^G^	0.32	2.73 ^J,K^	0.20
ZEN II	19.23 ^E^	2.11	12.26	0.67	18.28 ^L^	2.73
ZEN III	12.70	1.36	8.60	1.04	10.46	0.29

The results are expressed as means ± SEM (standard error of the mean). Statistically significant differences: 14 day, ^A^ Control Group vs. ZEN II, *p* < 0.0001, ^B^ Control Group vs. ZEN III, *p* < 0.0001, ^C^ ZEN I vs. ZEN II, *p* < 0.0001, ^D^ ZEN I vs. ZEN III, *p* < 0.001, ^E^ ZEN II vs. ZEN III, *p* < 0.001; 28 day, ^F^ Control Group vs. ZEN II, p < 0.0001, ^G^ ZEN I vs. ZEN II, *p* < 0.001; 42 day, ^H^ Control Group vs. ZEN II, *p* < 0.0001, ^I^ Control Group vs. ZEN III, *p* < 0.01, ^J^ ZEN I vs. ZEN II, *p* < 0.0001, ^K^ ZEN I vs. ZEN III, *p* < 0.0001, ^L^ ZEN II vs. ZEN III, *p* < 0.0001.

**Table 19 toxins-12-00350-t019:** Changes in proline content in the ileum. Experimental group and ZEN concentration: Control Group, ZEN I—5 μg/kg BW, ZEN II—10 μg/kg BW, ZEN III—15 μg/kg BW.

Group	Experimental Day					
	14		28		42	
	mean	SEM	mean	SEM	mean	SEM
Control	6.39	0.56	6.32 ^B^	0.73	5.04 ^E,F,G^	0.52
ZEN I	3.76 ^A^	0.49	2.51 ^C,D^	0.33	1.27 ^H,I^	0.16
ZEN II	8.29	1.19	5.77	0.36	10.76	1.32
ZEN III	6.57	0.83	6.37	1.08	10.21	0.98

The results are expressed as means ± SEM (standard error of the mean). Statistically significant differences: 14 day, ^A^ ZEN vs. ZEN II, *p* < 0.001; 28 day, ^B^ Control Group vs. ZEN I, *p* < 0.01, ^C^ ZEN I vs. ZEN II, *p* < 0.05, ^D^ ZEN I vs. ZEN III, *p* < 0.01; 42 day, ^E^ Control Group vs. ZEN I, *p* < 0.01, ^F^ Control Group vs. ZEN II, *p* < 0.0001, ^G^ Control Group vs. ZEN III, *p* < 0.0001, ^H^ ZEN I vs. ZEN II, *p* < 0.0001, ^I^ ZEN I vs. ZEN III, *p* < 0.0001.

**Table 20 toxins-12-00350-t020:** Declared composition of the complete diet.

Parameters	Composition Declared by the Manufacturer (%)
Wheat	55
Barley	22
Soybean meal	16
Wheat bran	4.0
Chalk	0.3
Zitrosan	0.2
Vitamin–mineral premix ^1^	2.5

^1^ Composition of the vitamin–mineral premix per kg: vitamin A—500.000 IU; iron—5000 mg; vitamin D^3^—100.000 IU; zinc—5000 mg; vitamin E (alpha-tocopherol)—2000 mg; manganese—3000 mg; vitamin K—150 mg; copper (CuSO_4_ ˑ 5H_2_O)—500 mg; vitamin B_1_—100 mg; cobalt—20 mg; vitamin B_2_—300 mg; iodine—40 mg; vitamin B_6_—150 mg; selenium—15 mg; vitamin B_12_—1500 µg; L-lysine—9.4 g; niacin—1200 mg; DL-methionine+cystine—3.7 g; pantothenic acid—600 mg; L-threonine—2.3 g; folic acid—50 mg; tryptophan—1.1 g; biotin—7500 µg; phytase+choline—10 g; ToyoCerin probiotic+calcium—250 g; antioxidant+mineral phosphorus and released phosphorus—60 g; magnesium—5 g; sodium; calcium—51 g.

**Table 21 toxins-12-00350-t021:** ELISA kits used for the determination of cytokine concentrations in porcine tissues. IL: interleukin, TGF: transforming growth factor, TNF: tumor necrosis factor.

Antigen	ELISA Test and Catalogue Number	Manufacturer, Country	Assay Range pg/mL
IL-1α	Porcine IL-1 alpha/IL-1F1 DuoSet ELISA, DY680	R&D systems, Minneapolis, MN, USA	93.8–6000 Intra-assay CV < 2.92% Inter-assay CV < 4.21%
IL-1β	Porcine IL-1 beta/IL-1F2 DuoSet ELISA, DY681	R&D systems, Minneapolis, MN, USA	62.5–4000 Intra-assay CV < 1.1% Inter-assay CV < 3.2%
IL-2	Porcine IL-1 alpha/IL-1F1 DuoSet ELISA, DY652	R&D systems, Minneapolis, MN, USA	46.9–3000 Intra-assay CV < 4% Inter-assay CV < 5.6%
IL-4	Porcine IL-4 DuoSet ELISA, DY654	R&D systems, Minneapolis, MN, USA	156.0–10,000 Intra-assay CV < 5% Inter-assay CV < 6.69%
IL-6	Porcine IL-6 DuoSet ELISA, DY686	R&D systems, Minneapolis, MN, USA	125.0–8000 Intra-assay CV < 1.98% Inter-assay CV < 5.76%
IL-8	Porcine IL-8/CXCL8 DuoSet ELISA, DY535	R&D systems, Minneapolis, MN, USA	125.0–8000 Intra-assay CV < 7.3% Inter-assay CV < 8%
IL-10	Porcine IL-10 DuoSet ELISA, DY693B	R&D systems, Minneapolis, MN, USA	23.4–1500 Intra-assay CV < 3% Inter-assay CV < 4.64%
IL-12/23p40	Porcine IL-12/IL-23 p40 DuoSet ELISA, DY912	R&D systems, Minneapolis, MN, USA	78.1–5000 Intra-assay CV < 3.67% Inter-assay CV < 4.25%
TNFα	Porcine TNF-alpha DuoSet ELISA	R&D systems, Minneapolis, MN, USA	31.2–2000 Intra-assay CV < 5.11% Inter-assay CV <5.24%
IFN-γ	Porcine IFN-gamma DuoSet ELISA, DY985	R&D systems, Minneapolis, MN, USA	62.5–4000 Intra-assay CV < 3.4% Inter-assay CV < 4.6%
TGFβ	TGF beta-1 Multispecies Matched Antibody Pair, CHC1683	ThermoFisher Scientific, Waltham, MA, USA	62.5–4000 Intra-assay CV < 2.9% Inter-assay CV < 5%

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
