# Peer review of "The Effect of Zearalenone on the Cytokine Environment, Oxidoreductive Balance and Metabolism in Porcine Ileal Peyer’s Patches"

_toxins, 2020, doi:10.3390/toxins12060350_

Round 1

Reviewer 1 Report

The authors present a well constructed and very interesting manuscript concerning the effects of a well-known and well-studied mycotoxin, zearalenone. 

I have some comments: 

1. Two important studies were published before by Zielonka et al (10.3390/toxins7083210 and 10.3390/toxins12030144). Despite of the high similarity observed between these two articles published and the present manuscript, the results presented and the common area of interest, these two articles were not cited. Why?

2. Concerning the methods used to evaluate the concentrations of some markers, why did not the authors choose a more sensitive and precise method of analysis, such as a chromatographic method?

3. Lines 445-446: In my opinion, no assumption can be launched in a scientific article without being based on concrete results, even if it is a logical deduction / intuition. Thus, I recommend to exclude this phrase.

4. The "References" section is too large, and some authors are over-cited (e.g. 15 articles of Obremski K are cited). I recommend to reduce these section, excluding some general information included in the "Discussion" section. Also, I recommend a balanced and correct citation strategy.

Author Response

Response to Reviewer 1 Comments.

  1. Two important studies were published before by Zielonka et al (10.3390/toxins7083210 and 10.3390/toxins12030144). Despite of the high similarity observed between these two articles published and the present manuscript, the results presented and the common area of interest, these two articles were not cited. Why?

I agree with the Reviewer that the mentioned papers are thematically close to the study described in this manuscript. I am certain that the lack of their citation was a result of overlooking. The quoted articles have been included in the improved version of the manuscript respectively in the lines 574 and 590.

  1. Concerning the methods used to evaluate the concentrations of some markers, why did not the authors choose a more sensitive and precise method of analysis, such as a chromatographic method?

In the research done the analyses were made with the spectrophotometric and ELISA  methods. Certainly, I agree with the Reviewer that some part of the parameter analysis could have been done with the HPLC methods. Unfortunately, at present we have not got experience in use of these methods.

  1. Lines 445-446: In my opinion, no assumption can be launched in a scientific article without being based on concrete results, even if it is a logical deduction / intuition. Thus, I recommend to exclude this phrase.

I agree with the Reviewer that this fragment of the conclusion is not based on the results obtained but rather on the discussion. Therefore, following thee suggestion of the Reviewer the fragment was deleted.  In this version the conclusion is presented in lines 423 and 431.

  1. The "References" section is too large, and some authors are over-cited (e.g. 15 articles of Obremski K are cited). I recommend to reduce these section, excluding some general information included in the "Discussion" section. Also, I recommend a balanced and correct citation strategy.

In accordance with the Reviewer’s suggestions an overview of the discussion and reference section was made. In the previous version the number of citations amounted to 82. Presently, it stands at 62. The share of the quotations of the articles by Obremski K. as the main author was reduced to 5. The other articles with his co-authorship are the result of the studies conducted in a team and are worth citing. In the present version over 50 quotations pertain the last 10 years. The deleted references were listed in the lines 557, 560, 568, 574, 579, 568, 591, 599, 602, 605, 610, 622, 627, 640, 645, 656, 671, 712, 716, 731.

In comparison to the reviewed version, the present version of the manuscripts lacks the fragments in the lines 73-87 and 268-269, whereas the lines 440-450 were re-edited.

Reviewer 2 Report

In my opinion the ms. "The effect of zearalenone on the cytokine environment, oxidoreductive balance and metabolism in porcine ileal Peyer’s patches" is of very correct trial setting, precise analytical approach and adequate data analysis.

My only concern would be the use of multiple graphs which are overloaded with horizontal comparison bars. I can accept this as well but tables with uppercase indices are more common.

Checking Table 1 data, the 4 main constituent of the diet are in summary more than 100%.

Please check this.

Soybean meal 16%

Wheat 55%

Barley 22%

Wheat bran 40%

The section L73-L88 seems for me less relevant, I suggest to shorten or delete it. This minor comments are generally not influencing my opinion on the quality of this work.

I suggest its acceptance after minor corrections.

Author Response

Response to Reviewer 2 Comments.

  1. My only concern would be the use of multiple graphs which are overloaded with horizontal comparison bars. I can accept this as well but tables with uppercase indices are more common.

I am perfectly aware of the fact that it is more common to present results in the form of a table. Moreover I fully conscious of the implication that the clarity of the results’ assessment is influenced by their graphic presentation. The results presented in the manuscript in the bar form are an outcome of a subjective convince of the authors.

  1. Checking Table 1 data, the 4 main constituent of the diet are in summary more than 100%.

The mistake in Table 1. is very obvious. The composition should certainly be made up of 100%. The error issues from the fact that in the item of wheat bran mistakenly was given the share of 40 %, instead of 4.0 %.

  1. The section L73-L88 seems for me less relevant, I suggest to shorten or delete it. This minor comments are generally not influencing my opinion on the quality of this work.

After a thorough analysis of the Reviewer’s comments it was decided not to shorten this fragment but to remove it in its entirety.

Round 2

Reviewer 1 Report

The authors present a improved version of the manuscript. I really appreciate the changes processed. Thus, in my opinion, the manuscript can be published in the present form. 

Author Response

In response to your review to the paper entitled “The effect of zearalenone on the cytokine environment, oxidoreductive balance and metabolism in porcine ileal Peyer’s patches”. Authors: Kazimierz Obremski, Wojciech Trybowski, Paweł Wojtacha, Magdalena Gajęcka, Józef Tyburski and Łukasz Zielonka, I inform kindly that as suggested, all figures have been replaced by tables.

Sincerely